# PRRSV Non-Structural Proteins Orchestrate Porcine E3 Ubiquitin Ligase RNF122 to Promote PRRSV Proliferation

**DOI:** 10.3390/v14020424

**Published:** 2022-02-18

**Authors:** Ruiqi Sun, Yanyu Guo, Xiaoyang Li, Ruiqiao Li, Jingxuan Shi, Zheng Tan, Lilin Zhang, Lei Zhang, Jun Han, Jinhai Huang

**Affiliations:** 1School of Life Sciences, Tianjin University, Tianjin 300072, China; srq_77@tju.edu.cn (R.S.); 146182@tju.edu.cn (Y.G.); 1019226002@tju.edu.cn (X.L.); liruiqiao@tjutcm.edu.cn (R.L.); shijingxuan@tju.edu.cn (J.S.); tz1947451114@gmail.com (Z.T.); lyliazhang@tju.edu.cn (L.Z.); zhanglei@tju.edu.cn (L.Z.); 2College of Veterinary Medicine, China Agricultural University, Beijing 100083, China

**Keywords:** RNF122, porcine reproductive and respiratory syndrome virus (PRRSV), nsp1, nsp4, nsp, ubiquitination

## Abstract

Ubiquitination plays a major role in immune regulation after viral infection. An alternatively spliced porcine E3 ubiquitin ligase RNF122 promoted PRRSV infection and upregulated in PRRSV-infected PAM cells was identified. We characterized the core promoter of RNF122, located between −550 to −470 bp upstream of the transcription start site (TSS), which displayed significant differential transcriptional activities in regulating the transcription and expression of RNF122. The transcription factor HLTF was inhibited by nsp1α and nsp7 of PRRSV, and the transcription factor E2F complex regulated by nsp9. Together, they modulated the transcription and expression of RNF122. RNF122 could mediate K63-linked ubiquitination to raise stability of PRRSV nsp4 protein and thus promote virus replication. Moreover, RNF122 also performed K27-linked and K48-linked ubiquitination of MDA5 to degrade MDA5 and inhibit IFN production, ultimately promoted virus proliferation. In this study, we illustrate a new immune escape mechanism of PRRSV that enhances self-stability and function of viral nsp4, thus, regulating RNF122 expression to antagonize IFNα/β production. The present study broadens our knowledge of PRRSV-coding protein modulating transcription, expression and modification of host protein to counteract innate immune signaling, and may provide novel insights for the development of antiviral drugs.

## 1. Introduction

Porcine reproductive and respiratory syndrome (PRRS) has been a major threat to the worldwide swine industry even since its first outbreak in late 1980s in the continents of both Europe and North America [1,2]. This disease often causes reproductive failure in sows and respiratory problems in pigs of all ages, leading to colossal economic losses, but no effective vaccines or anti-viral drugs are currently available [3]. The etiological agent is PRRS virus (PRRSV), an enveloped, positive-stranded RNA virus in the family Arteriviridae of the order Nidovirales. PRRSV has a genomic size of about 15 kb that contains at least 10 open reading frames (ORFs) [4,5]. The 3’ terminal ORFs2–7 code for structural proteins (e.g., GP2a, E, GP3, GP4, GP5, M and N, etc.) [6,7], whereas ORF1a and ORF1b account for nearly 75% of the 5’-end of the genome and encode viral two important viral replicase polyproteins (pp1a and pp1ab) that are further processed into at least 16 nonstructural replicase proteins (nsps). These nsps then come together to form viral replication and transcription complex (RTC) for genome replication and subgenomic RNA synthesis. Meanwhile, some nsps (e.g., nsp1, nsp2, nsp4, nsp9, nsp11, etc.) are critical in antagonizing host innate immunity [8].

Post-translational modification (PTM) is an important mechanism employed by eukaryotic cells and viruses to regulate the activities of proteins, ranging from metabolism, cell localization, signal transduction to protein stability, and so on [9,10]. Currently, there are more than 200 different types of PTMs (e.g., phosphorylation, ubiquitination, acetylation, glycosylation, methylation, SUMOylation, etc.) that have been discovered [11,12,13]. Of them, ubiquitination plays a crucial role in almost all cellular functions and can occur on any of the seven lysine residues of ubiquitin (K6, K11, K27, K29, K33, K48 and K63) or the N-terminal methionine (M1), resulting in eight different types of linkages to perform diverse functions [14]. For example, K48-linked ubiquitination usually regulates the turnover of cellular or viral proteins, whereas the K63-linked ubiquitination is often associated with signaling transduction cascade by regulating protein conformation. These two PMTs account for about 80% of all ubiquitin chain modifications in mammalian cells [15].

How PRRSV exploits ubiquitin modification system to achieve its maximal replication efficiency and to realize immune evasion has been subject to intensive investigations. The PRRSV nsp2 and nsp11 have been shown to possess deubiquitinating activities; they target not only cellular proteins such as RIG-I and MDA-5, but also promote the stabilization of viral proteins [16,17]. In addition, PRRSV targets host restriction factors such as CH25HC for degradation via ubiquitin proteasomal system [18]. Recent studies have shown that PRRSV utilizes E3 ubiquitin ligase ASB8 to increases the stabilization of nsp1α, which in turn inhibits NF-κB activity by targeting the linear ubiquitin chain assembly complex and also SLA-I for proteasome degradation [19,20]. Currently, how PRRSV interacts with cellular ubiquitin system to promote viral replication has remained poorly understood.

In this study, we looked into the role of ubiquitin E3 ligases of RNF family proteins by transcriptomic analysis. It has been reported that RNF122 plays a selective negative regulatory role in the antiviral innate response triggered by RIG-I by targeting the CARD domain of RIG-I and mediating the proteasomal degradation of RIG-I [21]. However, the role of RNF122 in other mammals has not been reported. Here, we identified RNF122 as a critical factor for PRRSV replication. During PRRSV infection, the viral replicase proteins nsp1a, nsp7 and nsp9 cooperatively upregulated the transcription of RNF122, which in turn mediated the degradation of pathogen pattern-recognition receptor MDA-5 through K27-linked and K48-linked ubiquitination, and promoted the stabilization of PRRSV nsp4 via K63-linked ubiquitination.

## 2. Materials and Methods

### 2.1. Cells and Virus

Porcine reproductive and respiratory syndrome virus (PRRSV) JXwn06 strain was preserved in our lab. Human embryonic kidney 293T cells (HEK293T) and HeLa cells were maintained in Dulbecco’s modified essential medium (DMEM) complemented with 10% fetal bovine serum (FBS) (HyClone Laboratories, Inc., Logan, UT, USA) and 1% penicillin/s treptomycin. Porcine pulmonary alveolar macrophages (PAM) cell line 3D4/21 (CRL-2843) was grown in RPMI-1640 medium complemented with 10% fetal bovine serum (FBS). All mammalian cell lines were grown at 37 °C with 5% CO_2_.

### 2.2. Antibodies and Reagents

Rabbit anti-RNF122 multiclonal antibody was prepared in the lab. Rabbit HLTF multiclonal antibody was purchased from ABclonal Technology (Wuhan, Hubei, China). Mouse anti-Flag monoclonal antibody and Mouse anti-Myc monoclonal antibody were purchased from Yeasen Biotech (Shanghai, China). Rabbit anti-HA polyclonal antibody was purchased from Cell Signaling Technology (Danvers, MA 01923, USA). Rabbit β-actin antibody was purchased from TransGen Biotech (Beijing, China). Mouse source Flag-tags beads were purchased from Abmart Co (Shanghai, China). Enzyme-labeled secondary antibodies: goat anti-mouse IgG HRP conjugate and goat anti-rabbit IgG HRP conjugate were purchased from SunGene Biotech (Tianjin, China).

### 2.3. Plasmid Construction

RNF122(GenBank accession number, XM_013984096.2) and MDA5 (NM_001100194.1) were cloned from porcine PBMC cells. Flag-RNF122, HA-RNF122 and Myc-RNF122 were cloned into pCMV2 modified plasmid and constructed by seamless cloning technology. Mutants of Myc-RNF122 were modified by our lab using Myc-RNF122 as the template. Flag-MDA5 was built by seamless cloning technology. The Flag tag was fused to PRRSV (JXwn06) gene, cloned into pCMV2 vectors, thereby constructing Flag-nsp1α, Flag-nsp1β, Flag-nsp4, Flag-nsp5, Flag-nsp7, Flag-nsp9, Flag-nsp10, Flag-nsp11, and Flag-N plasmids. The mutant plasmids of Flag-nsp4 and Flag-MDA5 were modified by our lab. Primers used in plasmid construction were given in Table 1. Flag-RIG-I was given by Tianjin Medical University.

### 2.4. Small RNA Interfering Assay

SiRNA targeting porcine RNF122 genes was synthesized by GenePharma Co (Shanghai). The RNF122 knock-down effect was evaluated according to the procedure as previously described [22]. The siRNA sequences were presented in Table 2.

### 2.5. Dual Luciferase Assays

pGL3-Basic and pGL3-Control were purchased from Promega Co. The promoter sequence of RNF122 (pTP1) was cloned using the pig genome extracted from PAM cells as a template. pTP1-Luc (Firefly luciferase reporter) was constructed by seamless cloning technology. The truncated promoter sequence plasmids pTP2-Luc, pTP3-Luc, pTP4-Luc, pTP5-Luc, pTP6-Luc, pTP7-Luc, pTP8-Luc, and pTP9-Luc were modified using pTP1-Luc as the template and primers were shown in Table 3.

HEK293T cells were seeded in 24-well plates and transfected with 100 ng pTP1-Luc, pTP2-Luc, pTP3-Luc, pTP4-Luc, pTP5-Luc, pTP6-Luc, pTP7-Luc, pTP8-Luc, or pTP9-Luc, respectively, and 20 ng RL-TK (Renilla luciferase control reporter) plasmid was used as control. The cells were inoculated with PRRSV at a multiplicity of infection (MOI = 0.5) or transfected with 100 ng PRRSV plasmids such as Flag-nsp1α, Flag-nsp1β, Flag-nsp4, Flag-nsp5, Flag-nsp7, Flag-nsp9, Flag-nsp10, Flag-nsp11, or Flag-N, respectively. Luciferase activity was measured using the supernatant of lysated cells 24 h later.

### 2.6. Quantitative Real-Time PCR

Total RNA from the cells was extracted using the RNAiso Plus reagent purchased from Takara, according to the manufacturer’s protocols. RNA (1 µg) was reverse-transcribed using reverse transcriptase (Yeasen Biotech), and SYBR Green Master Mix (Yeasen Biotech) was used for quantitative real-time RT-PCR analysis. qRT-PCR results were analyzed using the comparative cycle threshold (CT) method, and ABI 7500 Real-time PCR was used for numerical determination. The thermal cycling program consisted of 10 min at 95 °C followed by 45 cycles at 95 °C for 15 s, 55 °C for 30 s and 72 °C for 30 s. The primers are shown in Table 4.

### 2.7. Ubiquitination Assay

HEK293T cells were transfected with ubiquitin plasmids such as HA-Ub (WT), HA-Ub (K27), HA-Ub (K48), and HA-Ub (K63), respectively, together with RNF122 and nsp4/MDA5 plasmids. At 24 h after transfection, lysed cell supernatants were immunoprecipitated with Flag-specific antibodies. The ubiquitination analysis was then performed with an anti HA probe.

### 2.8. Immunofluorescence

HeLa cells or 3D4/21 cells were placed on cover slips in 12-well plates and co-transfected with the expression plasmids. Fixed with 4% paraformaldehyde for 30 min. Then, cells were permeated with PBS containing 0.3% tritonx-100 for 10 min, incubated with rabbit and mouse primary antibodies together for 1 h, washed with PBS for three times, and then incubated with fluorescence secondary antibody conjugate for 1 h. The anti-Flag, HA or Myc-tag fluorophores and co-location were detected by fluorescence confocal microscopy (UltraView Vox, PerkinElmer, Waltham, MA, USA).

### 2.9. Western Blot and Co-Immunoprecipitation Assays

HEK293T cells were seeded in six-well plates and co-transfect with Flag-tag plasmids. At 24 h, the inoculum was removed and the cells were first lysed for 15 min at 4 °C with 400 μL lysis buffer (50 mM Tris-HCl pH 7.5, 150 mM NaCl and 1% Nonidet-P40) supplemented with complete Mini Protease Inhibitor Cocktail (Roche Diagnostics, Mannheim, Germany). The soluble (supernatant) and insoluble (pellet) fractions were then obtained by centrifugation of the lysate for 15 min at 13,000 rpm. After pre-clearing, proteins of interest were precipitated by using Flag Beads (Abmart Shanghai Co., Ltd., Xuhui, Shanghai, China) to incubate at 4 °C for overnight. Precipitates were washed three times and supplemented with 50 μL buffer and then eluted by boiling in 10 μL SDS–PAGE sample buffer. Proteins were separated on denatured SDS-PAGE gels and transferred electrophoretic ally onto polyvinylidene fluoride (PVDF) membrane. Finally, proteins of interest were detected by enhanced chemiluminescence (GE Healthcare) and imaged using a chemiluminescence apparatus (Bio-Rad, Hercules, CA, USA).

### 2.10. Mass Spectrometry Analysis

Sample preparation, liquid chromatography–mass spectrometry, and data analysis were performed as described with a little modification [22]. Briefly, HEK293T cells were transfected with an expression vector for Flag-RNF122 or vector (FLAG only). After 24 h, the medium was discarded, 400 μL lysis buffer was added, cells were collected and broken, and placed on ice for 15 min for lysis. Fifty microliters of Flag beads were added into the upward purifier to rotate and combine overnight. We added 50 μL lysates to resuspend them and added 10 μL loading buffer and boiled them to ensure that IP samples. Proteins were separated using SDS-PAGE. Then the gel was stained with silver using a ProteoSilver Silver Stain Kit (Solarbio Life Sciences, Beijing, China). Each lane of the stained gel was cut, and individual gel slices were subject to in-gel digestion with trypsin, followed by analysis by mass spectrometry.

### 2.11. Statistical Analysis

Data were subjected to one-way analysis of variance (one-way ANOVA) and expressed as mean ± SEM. Prism 6.0 (GraphPad Software Inc. San Diego, CA, USA) was used to identify the statistical significance. All *p* values were two-sided, and statistical significance was assessed at *p* < 0.05. In addition, *p* < 0.01 and *p* < 0.001 were marked with one (*), two (**) and three (***) asterisks, respectively.

## 3. Results

### 3.1. Porcine E3 Ubiquitin Ligase RNF122 Was Up-Regulated after PRRSV Infection

Accumulating evidence suggests that E3 ubiquitin ligase plays an important role in viral infection [23]. Among them, the mechanism of RING domain family E3 ubiquitin ligase in innate immune signaling pathway has been extensively studied [24,25,26]. More than 2000 proteins were significantly changed in the transcriptome sequencing of PRRSV-infected PAM cells as we previously described [27]. A total of 26 homologous enzymes containing RING domain E3 ubiquitin ligases in the transcriptomic data were further identified and the significantly up-regulated RNF122 was screened for further study. (Figure 1a). The porcine RNF122 expression was further proven in PRRSV infected PAM (3D4/21) cells. The transcription level of RNF122 was significantly up-regulated at 12 h post PRRSV infection, and achieved the maximum at 24 h (Figure 1b). In addition, fluorescent photography confirmed that the expression level of RNF122 was also significantly up-regulated in PRRSV-infected cells (Figure 1c). All results showed that the transcription and protein levels of RNF122 were significantly upregulated in PRRSV-infected PAM cells.

### 3.2. The Core Promoter Region of Porcine RNF122 Was Identified

Promoter is an important part of DNA sequence recognized, bound and initiated by RNA polymerase [28]. To understand the molecules that promote the upregulation of RNF122 post viral infection, the promoter sequences in the porcine genome were further analyzed. The potent promoter sequences located at −1800 bp to 0 bp upstream of the porcine RNF122 gene initial codons were predicted through NCBI (Available online: https://www.ncbi.nlm.nih.gov/gene/100511414, accessed on 2 November 2019) analysis and further cloned from 3D4/21 cells. After that, the pTP1-Luc report plasmid was constructed by inserting the fragment into pGL3-Basic plasmid, and enhanced promoter activity after PRRSV infection was further verified (Figure 2c). The core promoter, usually a 50–200 bp DNA fragment, is a key element in the regulation of gene activity [29]. To further determine the core promoter region of RNF122, eight truncated mutants (named pTP2-Luc, pTP3-Luc, pTP4-Luc, pTP5-Luc, pTP6-Luc, pTP7-Luc, pTP8-Luc and pTP9-Luc, respectively) were constructed (Figure 2a) and the promoter activities were detected. Fluorescent high enzyme activity of pTP6-Luc indicating that the core promoter region of RNF122 positioning in −550 bp to −470 bp region (Figure 2b), and the promoter activity of pTP6-Luc was also significantly enhanced after PRRSV infection (Figure 2c).

### 3.3. E2F Complex and HLTF Were Identified as Key Transcription Factors in Porcine RNF122

The initiation or inhibition of transcription of genes mainly occurs after the binding of transcription factors to the promoter sequence [30]. To determine which transcription factors bind to the core promoter region and play a role in regulating transcription, the online tool (Available online: http://jaspar.genereg.net/, accessed on 4 January 2020) was used to predict the transcription factor binding sites and the potent transcription factors which binding these core promoter region, and the potent transcription factors E2F, HLTF, FOXC1 and NFIC were predictive involved (Table 5). To confirm the transcription activity of these predicted transcription factors, three pTP6-Luc mutation plasmids targeted different transcription factor binding regions were constructed taken into account their potent binding sites (Figure 3a). Significantly reduced promoter activity results of the pTP6-M1-Luc mutant plasmid indicated that the E2F complex positively regulates the transcription of RNF122 gene. On the contrary, the promoter activity of pTP6-M2-Luc and pTP6-M3-Luc was markedly up-regulated, indicating that HLTF, NFIC and FOXC1 negatively regulated the transcription of RNF122 gene (Figure 3b). To further clarify which transcription factors play a role in regulating RNF122 expression, the gene expression level of these transcription factors in PRRSV infected cells was determined by real time-qPCR. We found that after PRRSV infection, the transcription level of HLTF was significantly down-regulated, which was different from missing detection signal of NFIC and stable expression level of FOXC1, indicating that PRRSV infection down-regulated the HLTF expression and further promoted the transcription of RNF122 gene (Figure 3c). Interestingly, we found that PRRSV infection did not impact the transcription level of E2F complex, but the E2F complex was previously reported as a positively regulated transcription factor, and RB protein can be bind to the E2F complex and inhibit E2F activity [31]. We also discovered that the transcription level of RB was down-regulated after PRRSV infection. This suggested that PRRSV can further reduce the activity of E2F complex and promote the transcription of RNF122 by down-regulating the transcription level of RB (Figure 3c). So, we identified the E2F complex and HLTF as key transcription factors in regulating the transcription of RNF122 gene.

### 3.4. Transcriptional Regulations of RNF122 Were Mediated by PRRSV nsp1α, nsp7 and nsp9

Viral proteins may be involved in regulation of host gene expression by engaging in dialogue with host transcription factors [32]. To find potent PRRSV viral protein involved in impacting the transcription of RNF122 gene, eukaryotic expression plasmids of PRRSV genes were manufactured and co-transfected with pTP6-Luc. The results showed that PRRSV nsp1α, nsp7 and nsp9 up-regulated the promoter activity of pTP6-Luc significantly (Figure 4a). It has been further reported that the nsp9 protein of PRRSV can interact with RB protein and further impair the expression of RB protein [32]. To further verify the effect of nsp9 on the transcription level of RNF122, the co-transfected cells were performed with nsp9 and pTP6-Luc plasmids, the results were verified by decreasing RB transcription level (Figure 4c) and an enhancing pTP6-Luc promoter activity (Figure 4b), In addition, transcription level of RNF122 was significantly increased (Figure 4c) after nsp9 plasmid transfection. These results reported that PRRSV could down-regulate RB expression through nsp9 and further promote the transcription of RNF122. On the other hand, as we previously verified results of up-regulated promoter activity of pTP6-Luc by PRRSV nsp1α and nsp7, and HLTF is regulated as a key transcription factor of RNF122 gene, we need to define whether nsp1α and nsp7 impact the transcription of RNF122 by regulating the expression of HLTF. Therefore, we transfected cells with nsp1α or nsp7, respectively, and found that the transcription (Figure 4d) and protein expression level (Figure 4e) of HLTF were significantly down-regulated post-transfection. This stated that PRRSV promotes the transcription of RNF122 gene by down-regulating the expression of HLTF through viral nsp1α and nsp7.

### 3.5. Porcine RNF122 Promoted Viral Replication of PRRSV

Many viruses can antagonize the intrinsic cellular antiviral response by adjusting expression and modification of endogenous protein. Thus, promoting the replication of the virus [19,22]. To understand the antagonistic effect of RNF122 on cellular intrinsic antiviral proliferation, the proliferation of PRRSV was detected by over expressing and interfering RNF122. As the largest viral protein of PRRSV, nsp2 protein is a direct result of virus replication [33]. Nucleocapsid (N) protein is a polyphosphorylated protein whose phosphorylation contributes to viral replication and virulence [34]. The transcription levels of N and nsp2 gene of PRRSV were significantly up-regulated under over expression of RNF122 (Figure 5a,b), and the protein expression of nsp2 was also significantly increased (Figure 5c). The transcription levels of N and nsp2 of PRRSV were significantly down-regulated (Figure 5d) in the case of RNF122 siRNA transfection (Table 2), and the expression level of the corresponding nsp2 protein was also significantly down-regulated (Figure 5f). Viral copies of PRRSV increased after transfection with RNF122. However, after RNF122 was interfered, the viral copies of PRRSV were reduced (Figure 5g). Virus titers showed the same effect (Figure 5h).

### 3.6. Interaction between Porcine RNF122 and PRRSV nsp4

PRRSV non-structural proteins can impact viral proliferation by interacting directly with host protein or mediating innate immune signaling pathways [19,22]. RNF122 plasmid was co-transfected with expression vectors that encode PRRSV proteins, and the immunoprecipitation experiment showed that RNF122 interacted with nsp4 (Figure 6a) and co-localization relationship (Figure 6b) was also pointed out in confocal fluorescence microscope experiment. The RNF122 can obviously promote expression of nsp4 with dose-dependent manner (Figure 6c). Porcine RNF122 which displays the relying effect of doses, can engage in dialogue with nsp4 of PRRSV and prompt PRRSV proliferation.

### 3.7. RNF122 Performed K63-Linked Ubiquitination Lysine of PRRSV nsp4 at Position 170

The ubiquitination modification between RNF122 and nsp4 was further performed, and the K63-linked ubiquitination of nsp4 by RNF122 was confirmed (Figure 7a,b and Appendix A). The main role of K63-linked ubiquitination is to stabilize proteins and promote protein expression [35]. In order to further find specific ubiquitination sites of nsp4 activated by RNF122, five lysine mutants of nsp4 were constructed based on references and co-transfected with RNF122 [36]. The disappeared ubiquitination determined in nsp4 K170R mutant, indicated that RNF122 could ubiquitinate lysine at position 170 of nsp4 (Figure 7c). These results suggest that RNF122 interacts with nsp4 of PRRSV, and RNF122 can perform K63-linked ubiquitination to 170th lysine in nsp4.

### 3.8. Porcine RNF122 Negatively Regulated Type I Interferon Signaling Pathway

E3 ubiquitin ligases play an important role in the regulation of type I interferon signaling pathway [37,38]. The role of RNF122 in antagonizing the type I interferon signaling pathway is difficult to say. The transcription level of IFN-β and NF-κB were significantly down-regulated (Figure 8a) under over expression of RNF122 (Figure 8c). From another perspective, an increasing transcription level of IFN-β (Figure 8d) and NF-κB (Figure 8e) were observed in the RNA interference treatment of RNF122 (Figure 8f,g). The results stated that RNF122 negatively regulate the type I interferon signaling pathway.

### 3.9. Interaction between Porcine RNF122 and MDA5 Molecules

To further identify the potent interacting proteins with RNF122 which involve in type I interferon signaling pathways, the potent interaction proteins with RNF122 were screened by anti-Flag-beads from RNF122-tag plasmid transfection cells, and the mass spectrometry analysis was performed to find the potent interaction proteins (Figure 9a). A total of 356 interaction proteins of RNF122 were determined and the high abundance of MDA5 was screened for further analysis (Table 6). The previous literature has reported that RNF122 interacts with RIG-I and negatively regulates type I interferon signaling pathway by ubiquitination of RIG-I [21]. In order to further test the interaction between RNF122 and MDA5, MDA5 and RNF122 were co-transfected into HEK293T cells. Co-immunoprecipitation result showed that RNF122 interacts with MDA5 (Figure 9b), and co-localization identified between them by fluorescence labeling detection (Figure 9c). Three truncated domains of MDA5 (MDA5-N, MDA5-M and MDA5-C) and two truncations of RNF122 (TM and RING region of RNF122) were constructed for further to identify co-interaction region of them. Interaction between RNF122 and MDA5 was determined in MDA5-N (Figure 9d) and RNF122 (TM) region (Figure 9e). All in all, RNF122 and MDA5 interact with each other, mainly dependent on the N-terminal CARD domain of MDA5 and the TM domain of RNF122.

### 3.10. Porcine RNF122 Performed K27-Linked and K48-Linked Ubiquitination to MDA5

RNF122 is an E3 ubiquitin ligase that negatively regulates type I interferon signaling and interacts with MDA5. It was previously reported that RNF122 negatively regulated type I interferon signaling pathway by K48-linked ubiquitination to RIG-I [21], and another E3 ubiquitin ligase TRIM40 negatively regulated type I interferon signaling pathway by K27-linked and K48-linked ubiquitination to MDA5 and RIG-I [39]. The ubiquitination of MDA5 by RNF122 was determined, the expression of MDA5 gradually decreased with the increasing RNF122 plasmid transfection by co-transfected RNF122 and MDA5 plasmids into HEK293T cells, meanwhile no significant expression change of MDA5 after the MG132 supplement treated cells (Figure 10a), the results indicated that RNF122 prompted the degradation of MDA5 by the proteasome pathway. The K27-linked (Figure 10c) and K48-linked (Figure 10b) ubiquitination to MDA5 performed by RNF122 were further identified (Appendix A). The K27-linked and K48-linked ubiquitination are primarily responsible for protein degradation [40]. In order to further confirm which lysine site of MDA5 is activated by RNF122, eight mutations in MDA5 were constructed (Figure 10d) and co-transfected with RNF122 plasmid. The results showed that the significantly reduced ubiquitination in MDA5 (K68R) and MDA5 (K137R) mutants, and the ubiquitination disappeared when simultaneous mutation of the two sites was observed (Figure 10e). Then, to further define which lysine sites of MDA5 modified by K27-linked or K48-linked ubiquitination, the Ub-K27 and Ub-K48 plasmids were co-transfected with the two MDA5 lysine mutants, respectively. It was found that the K27-linked and K48-linked ubiquitination can be carried on both lysine sites of MDA5 (Figure 10f,g). These results indicated that RNF122 could perform K27-linked and K48-linked ubiquitination to K68 and K137 of MDA5, and then negatively regulate the type I interferon signaling pathway.

## 4. Discussion

Porcine Reproductive and respiratory syndrome (PRRS) causing significant economic losses to the pig industry worldwide [41]. PRRSV infection causes a strong immune response, and destroys the innate immune system [42], and impairs adaptive immune response [43]. Viral infections trigger an antiviral response in host cells, often known as an “innate immune response” [44]. An important class of molecules that activate innate antiviral responses are pattern recognition receptors, which recognize pathogen-related molecular patterns and lead to the activation of IFN after a series of signal transduction reactions, thereby exerting antiviral activity [45]. A toll-like receptor (TLR) and RIG-like receptor (RLR) are considered to be key sensors of PRRSV infection. Inhibition of the signal transduction and production of type I IFN are the principal features of PRRSV infection, which lead to persistent infection. PRRSV encodes several proteins, such as nsp1α, nsp1β, nsp2, nsp4, nsp11, and N that act as antagonists for the IFN signaling [46]. As evolution has progressed, viruses have been able to antagonize this response by using the body’s own proteins, such as nsp1 of PRRSV can degrade CREB binding protein (CBP) and inhibit the formation of enhanced corpuscles, thereby inhibiting the production of IFN [47]. The expression of PRRSV nsp2TF and nsp2N in transfected cells showed that they have ability to inhibit the innate immune response of cells [33]. In this process, post-translational modifications of proteins play an important role, especially phosphorylation and ubiquitination [23], which directly determine the direction of signal transduction.

E3 ubiquitin ligase plays a direct catalytic role in protein ubiquitination, and there is increasing evidence that viruses can use the catalytic activity of E3 ubiquitin ligase to target host and viral ubiquitination mechanisms [19,21,39]. One is that the virus encodes E3 ubiquitin ligase, which enhances its own replication by proteasomal degradation or inhibition of host immune proteins. The other is that the virus hijacks the host’s E3 ubiquitin ligase to interfere with the IFN-mediated antiviral response or regulate viral proliferation by ubiquitination or deubiquitination of viral proteins [22]. The E3 ubiquitin ligases are the important targets of immune evasion in viral infection.

In our study, the E3 ubiquitin ligase RNF122 up-regulated after PRRSV infection was screened, and its ubiquitination modification characters and regulation mechanisms were investigated. It has been reported that RNF122 plays a selective negative regulatory role in the antiviral innate response triggered by RIG-I by targeting the CARD domain of RIG-I and mediating the proteasomal degradation of RIG-I. However, the role of RNF122 in other mammals and the interaction between PRRSV with cellular ubiquitin system to promote viral replication has remained poorly understood. PRRSV nsp9 can target RB protein [32], an inhibitor of transcription factor complex E2F [31] were investigated as previously described. Therefore, nsp9 can promote the action of E2F. In the study, the important transcription factor E2F acted in the core promoter region of RNF122, and its transcriptional activity promoted by PRRSV nsp9 was verified, which further up-regulated the transcription of RNF122. In addition, another RNF122 transcription factor HLTF down-regulated by the nsp1α and nsp7 of PRRSV was also being identified, which promoted the transcription of RNF122 by attenuating the activity of repressing the RNF122 promoter. These results demonstrate that the transcription factor of RNF122 can be influenced by three PRRSV proteins, nsp1α, nsp7 and nsp9. However, the specific mechanisms of how the HLTF expression inhibited by PRRSV nsp1α and nsp7 in detail are not yet clear, which needs to be clarified in future research.

We also discovered that the RNF122 prompted PRRSV proliferation. On one hand, RNF122 directly interacts and K63-linked ubiquitination PRRSV nsp4 protein to stabilize the expression of nsp4. On the other hand, we found that RNF122 can target MDA5 to regulate the type I interferon signaling pathway. RNF122 performs K27-linked and K48-linked ubiquitination to MDA5 to prompt degradation of it, thereby inhibiting the IFN downstream signaling pathway and reducing the production of IFN, ultimately, promoting the proliferation of PRRSV. It was therefore suggested that PRRSV nsp4 could hijack RNF122 and enhance virus replication by promoting self-virus protein replication and inhibiting type I interferon signaling pathway.

In summary, we found PRRFV non-structural protein (nsp1α, nsp7, nsp9) play synergy effects to regulate E3 ubiquitin ligase RNF122 expression level by modulating expression of its three transcription factors. Moreover, RNF122 performs K63-linked ubiquitination to stabilize and enhance the function of nsp4 to promote the proliferation of PRRSV. The thirdly, RNF122 performs K27-linked and K48-linked ubiquitination to MDA5 to inhibit IFN production. A model in which PRRSV could antagonize the antiviral response and enhance its proliferation by regulation of the E3 ubiquitin ligase RNF122 was proposed. This is helpful for understanding the complex immune escape mechanisms of PRRSV infection and providing a potent drug target for prevention and therapy of the disease.

## Figures and Tables

**Figure 1 viruses-14-00424-f001:**
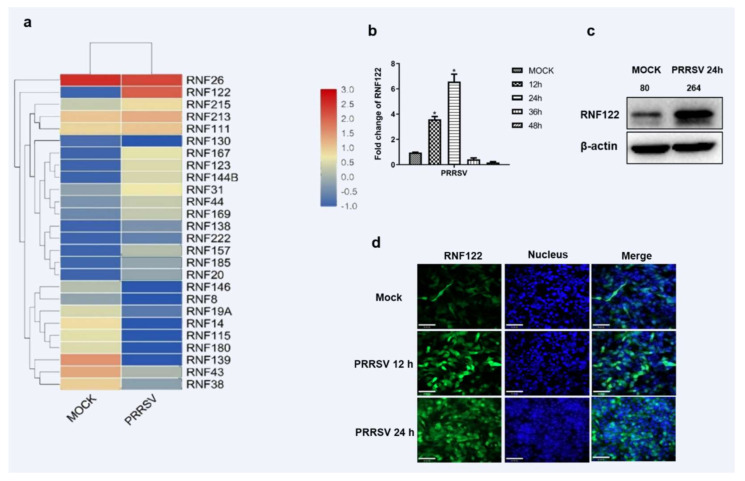
Porcine E3 ubiquitin ligase RNF122 was up-regulated after PRRSV infection. (**a**) Heat map of differential expression of E3 ubiquitin ligase in RING domain family after PRRSV infection. (**b**) 3D4/21 cells were infected with PRRSV (MOI = 0.5), and the cells were harvested at different time points for qRT-PCR detection to analyze the expression of RNF122. (**c**) 3D4/21 cells were infected with PRRSV (MOI = 0.5), and (**d**) the expression of RNF122 was observed by fluorescence microscopy at different time points. * *p <* 0.05 (analysis of two-way ANOVA attended by Bonferroni post-test). Data are representative of three independent experiments.

**Figure 2 viruses-14-00424-f002:**
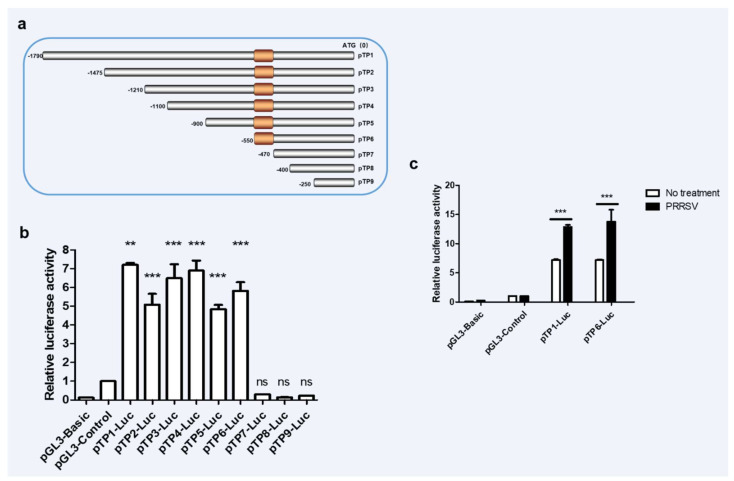
The core promoter region of porcine RNF22 was identified. (**a**) Schematic diagram of constitution of luciferase mutant plasmid. (**b**) HEK293 T cells were co-transfected with Luciferase mutant plasmid and RL-TK, luciferase activity was measured using the supernatant of lysate cells. (**c**) HEK293 T cells were co-transfected with pTP1-Luc, pTP6-Luc and RL-TK, infected with PRRSV (MOI = 0.5) or not, luciferase activity was measured using the supernatant of lysate cells. pGL3-Basic was the negative control and pGL3-Control was the positive control. ** *p* < 0.01; *** *p* < 0.001 ns, no significant difference (analysis of two-way ANOVA attended by Bonferroni post-test). Data are representative of three independent experiments.

**Figure 3 viruses-14-00424-f003:**
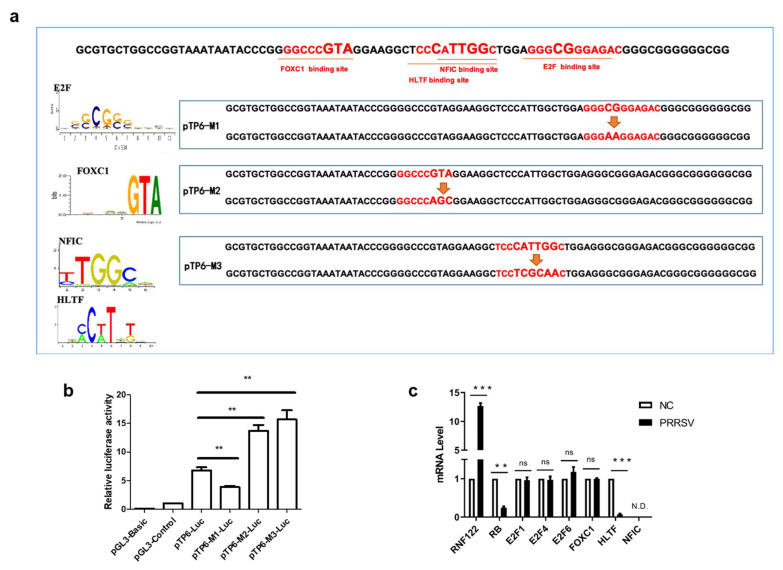
E2F complex and HLTF were identified as key transcription factors in porcine RNF122. (**a**) Schematic diagram of transcription factor binding site mutation in RNF122 core promoter region. (**b**) HEK293T cells were co-transfected with mutant plasmids of pTP6-Luc and RL-TK, luciferase activity was measured using the supernatant of lysate cells. pGL3-Basic was the negative control and pGL3-Control was the positive control. (**c**) 3D4/21 cells were infected with PRRSV (MOI = 0.5), and the cells were collected 24 h later for qRT-PCR detection to analyze the expression of transcription factors. ** *p* < 0.01; *** *p* < 0.001; ns, no significant difference; N.D., no determined. (analysis of two-way ANOVA attended by Bonferroni post-test). Data are representative of three independent experiments.

**Figure 4 viruses-14-00424-f004:**
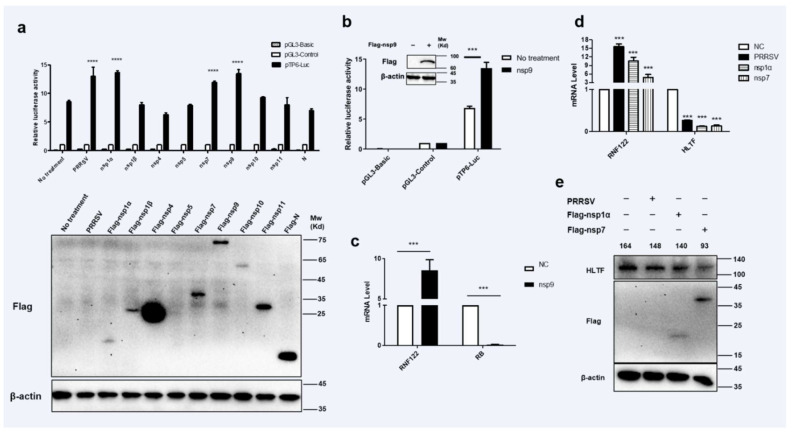
Transcriptional regulations of RNF122 were mediated by PRRSV nsp1α, nsp7 and nsp9. (**a**) HEK293T cells were co-transfected with pTP6-Luc and pRL-TK, and then transfected nsp1α, nsp1β, nsp4, nsp5, nsp7, nsp9, nsp10, nsp11, N plasmids, respectively, or infected with PRRSV (MOI = 0.5), luciferase activity was measured using the supernatant of lysate cells. pGL3-Basic was the negative control and pGL3-Control was the positive control. (**b**) HEK293T cells were co-transfected with pTP6-Luc and RL-TK, transfected with nsp9 or not, luciferase activity was measured using the supernatant of lysate cells. (**c**) 3D4/21 cells were infected with nsp9 or not, and the cells were collected 24 h later for qRT-PCR detection to analyze the expression of RNF122 and RB. (**d**) 3D4/21 cells were infected with PRRSV (MOI = 0.5), or transfected nsp1α, nsp7 or not, and the cells were collected 24 h later for qRT-PCR detection to analyze the expression of RNF122 and HLTF. (**e**) 3D4/21 cells were infected with PRRSV (MOI = 0.5), or transfected nsp1α, nsp7 or not, and the cells were collected 24 h later to analyze the protein expression of HLTF. *** *p* < 0.001; ***** p <* 0.0001 (analysis of two-way ANOVA attended by Bonferroni post-test). Data are representative of three independent experiments.

**Figure 5 viruses-14-00424-f005:**
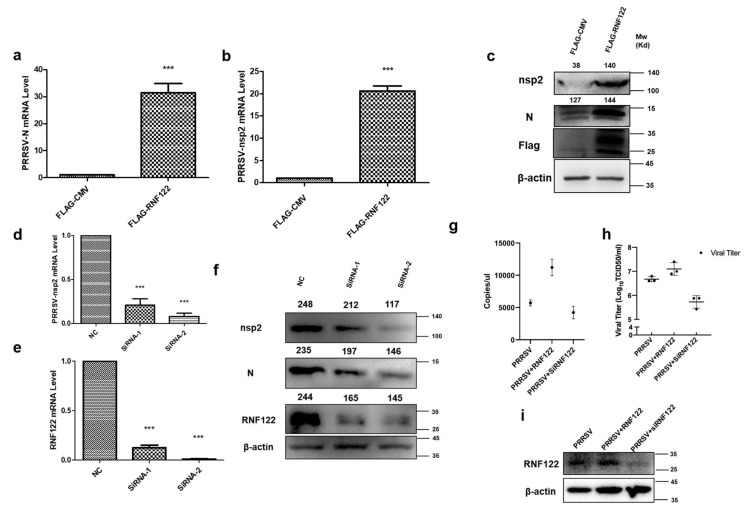
Porcine RNF122 promoted viral replication of PRRSV. (**a**,**b**) 3D4/21 cells were transfected with Flag-CMV or Flag-RNF122, 12 h later, cells were infected with PRRSV (MOI = 0.5). mRNA loads of PRRSV N and nsp2 were tested 12 h later by qRT-PCR. (**c**) 3D4/21 cells were transfected with Flag-CMV or Flag-RNF122, 12 h later, cells were infected with PRRSV (MOI = 0.5). PRRSV nsp2 were 12 h later tested by WB. (**d**,**e**) 3D4/21 cells were transfected with SiRNA or not; 12 h later, cells were infected with PRRSV (MOI = 0.5). mRNA loads of PRRSV nsp2 and RNF122 were tested 12 h later by qRT-PCR. (**f**) 3D4/21 cells were transfected with SiRNA or not; 12 h later, cells were infected with PRRSV (MOI = 0.5). PRRSV nsp2 was tested 12 h later by WB. (**g**) 3D4/21 cells were transfected with RNF122 or SiRNA; 12 h later, cells were infected with PRRSV (MOI = 0.5). PRRSV-N genes as an indicator to represent virus copy number and viral copies of PRRSV were tested 12 h later by qRT-PCR. (**h**) 3D4/21 cells were transfected with RNF122 or SiRNA; 12 h later, cells were infected with PRRSV (MOI = 0.5). TCID50 was calculated 48 h later by Reed Muench method. (**i**) 3D4/21 cells were transfected with RNF122 or SiRNA; 12 h later, cells were infected with PRRSV (MOI = 0.5). RNF122 was checked 12 h later by WB. *** *p* < 0.001 (analysis of two-way ANOVA attended by Bonferroni post-test). Data are representative of three independent experiments.

**Figure 6 viruses-14-00424-f006:**
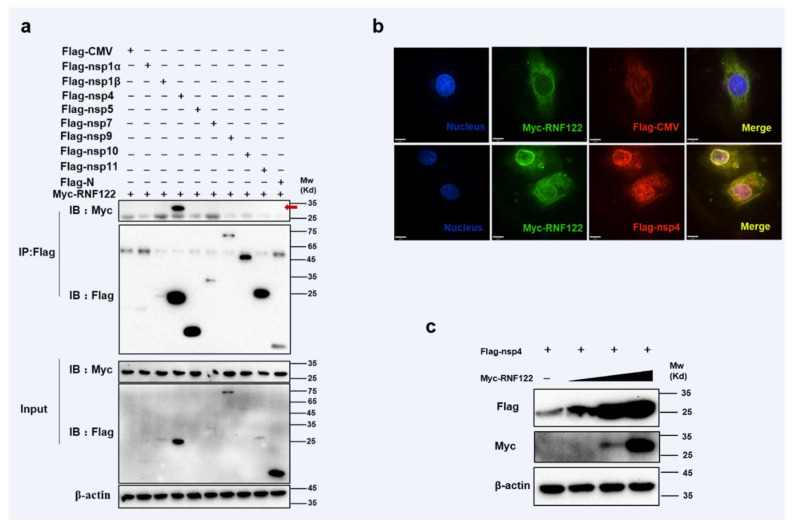
Interaction between porcine RNF122 and PRRSV nsp4. (**a**) HEK293T cells were co-transfected with Myc-RNF122 and Flag-nsp1α, Flag-nsp4, Flag-nsp5, Flag-nsp7, Flag-nsp9, Flag-nsp10, Flag-nsp11, Flag-N or Flag-CMV. Flag Beads were used for CO-IP 24 h after transfection and further detected by Western blotting with an anti-Flag antibody and an anti-Myc antibody, respectively. Red arrow represents interaction bands. (**b**) 3D4/21 cells were placed on cover slips in 12-well plates and co-transfected with the Myc-RNF122 and Flag-nsp4 or Flag-CMV, respectively. Fixed double-stained with a mouse anti-Myc mAb and a rabbit anti-Flag antibody, and attended by FITC-conjugated anti-mouse IgG (green) and PE-conjugated anti-rabbit IgG (red). Nuclei were stained with DAPI (blue). Fluorescence confocal microscopy (UltraView Vox, PerkinElmer, Waltham, MA, USA) was used to detect the co-location. (**c**) HEK293T cells were co-transfected with Flag-nsp4 (1 μg) and Myc-RNF122 (0 μg, 0.1 μg, 0.5 μg, 1 μg), 24 h later, the protein expression was tested by WB.

**Figure 7 viruses-14-00424-f007:**
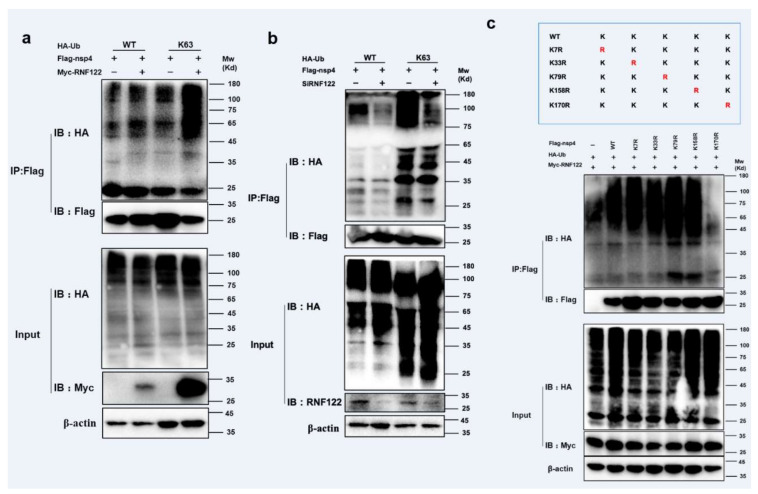
RNF122 performed K63-linked ubiquitination lysine of PRRSV nsp4 at position 170. (**a**) HEK293T cells were co-transfected with Myc-RNF122, Flag-nsp4 and HA-Ub (WT) or HA-Ub (K63). Flag Beads were used for CO-IP 24 h after transfection and further detected by Western blotting with an anti-HA, an anti-Flag antibody and an anti-Myc antibody, respectively. (**b**) HEK293T cells were co-transfected with SiRNF122, Flag-nsp4 and HA-Ub (WT) or HA-Ub (K63). Flag Beads were used for CO-IP 24 h after transfection and further detected by Western blotting with an anti-HA, an anti-Flag antibody and an anti-RNF122 antibody, respectively. (**c**) HEK293T cells were co-transfected with Myc-RNF122, HA-Ub (WT) and Flag-nsp4, Flag-nsp4(K7R), Flag-nsp4(K33R), Flag-nsp4(K79R), Flag-nsp4(K158R), Flag-nsp4(K170R) or not. Flag Beads were used for CO-IP 24 h after transfection and further detected by Western blotting with an anti-HA, an anti-Flag antibody and an anti-Myc antibody, respectively.

**Figure 8 viruses-14-00424-f008:**
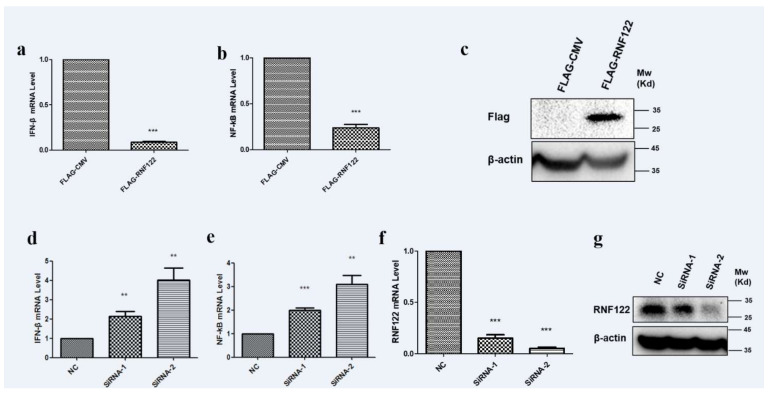
Porcine RNF122 negatively regulated type I interferon signaling pathway. (**a**,**b**) 3D4/21 cells were transfected with Flag-CMV or Flag-RNF122, 24 h later, mRNA loads of IFNβ and NF-κB were tested by qRT-PCR. (**c**) 3D4/21 cells were transfected with Flag-CMV or Flag-RNF122, 24 h later, RNF122 was tested by WB. (**d**–**f**) 3D4/21 cells were transfected with SiRNA or not, 24 h later, mRNA loads of IFN-β, NF-κB and RNF122 were tested by qRT-PCR. (**g**) 3D4/21 cells were transfected with SiRNA or not, 24 h later, RNF122 was tested by WB. ** *p* < 0.01; *** *p* < 0.001 (analysis of two-way ANOVA attended by Bonferroni post-test). Data are representative of three independent experiments.

**Figure 9 viruses-14-00424-f009:**
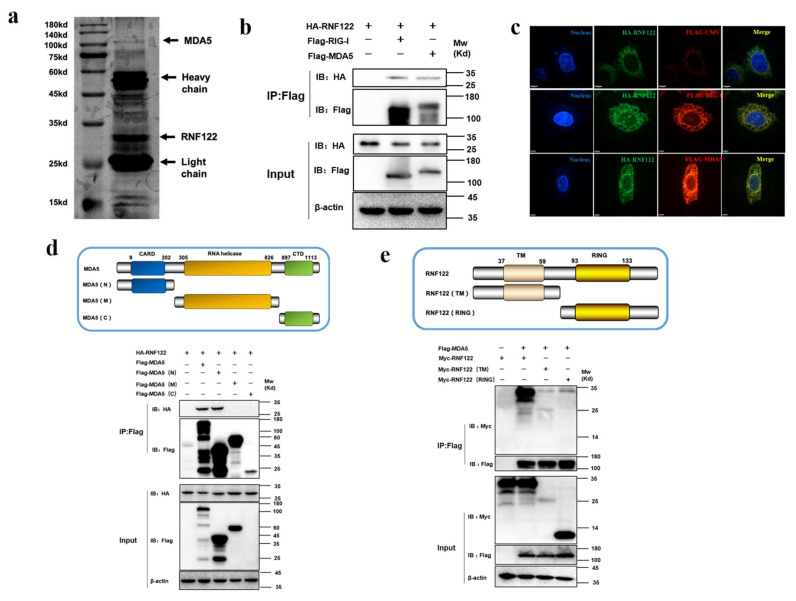
Interaction between porcine RNF122 and MDA5 molecules. (**a**) HEK293T cells were co-transfected with Flag-RNF122, 24 h after transfection, Flag Beads were used for protein enrichment, and further detected by Western blotting. Then the gel was stained with silver using a ProteoSilver Silver Stain Kit (Solarbio Life Sciences). (**b**) HEK293T cells were co-transfected with HA-RNF122 and Flag-RIG-I, Flag-MDA5 or Flag-CMV. Flag Beads were used for CO-24 h after transfection IP, and further detected by Western blotting with an anti-Flag antibody and an anti-HA antibody, respectively. (**c**) HeLa cells were placed on coverslips in 12-well plates and co-transfected with the HA-RNF122 and Flag-RIG-I, Flag-MDA5 or Flag-CMV, respectively. Fixed double-stained with a mouse anti-HA mAb and a rabbit anti-Flag antibody, and followed by FITC-conjugated anti-mouse IgG (green) and PE-conjugated anti-rabbit IgG (red). Nuclei were stained with DAPI (blue). Fluorescence confocal microscopy (UltraView Vox, PerkinElmer, Waltham, MA, USA) was used to detect the co-location. (**d**) HEK293T cells were co-transfected with HA-RNF122 and Flag-MDA5, Flag-MDA5 (N), Flag-MDA5 (M), Flag-MDA5 (C)or Flag-CMV. Flag Beads were used for CO-IP 24 h after transfection, and further detected by Western blotting with an anti-Flag antibody and an anti-HA antibody, respectively. (**e**) HEK293T cells were co-transfected with Flag-MDA5 and Myc-RNF122, Myc-RNF122 (TM), Myc-RNF122 (RING) or Myc-CMV. Flag Beads were used for CO-IP 24 h after transfection and further detected by Western blotting with an anti-Flag antibody and an anti-Myc antibody, respectively.

**Figure 10 viruses-14-00424-f010:**
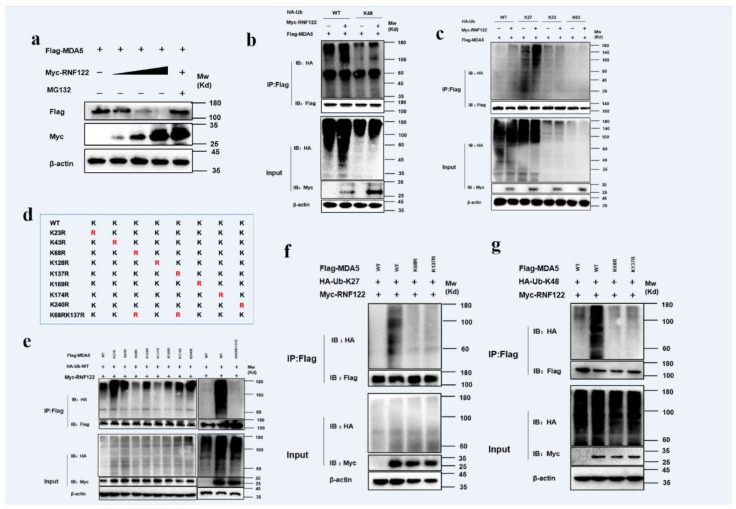
Porcine RNF122 performed K27-linked and K48-linked ubiquitination to MDA5. (**a**) HEK293T cells were co-transfected with Flag-MDA5 (1 μg) and Myc-RNF122 (0 μg, 0.5 μg, 1 μg, 2 μg), 12 h later, cells were infected with MG132 or not, 12 h later, the protein expression was tested by WB. (**b**,**c**) HEK293T cells were co-transfected with Myc-RNF122, Flag-MDA5 and HA-Ub (WT), HA-Ub (K48), HA-Ub (K2), HA-Ub (K33) or HA-Ub (K63). Flag Beads were used for CO-IP 24 h after transfection and further detected by Western blotting with an anti-HA, an anti-Flag antibody and an anti-Myc antibody, respectively. (**d**) Schematic of a mutant of Flag-MDA5. (**e**) HEK293T cells were co-transfected with Myc-RNF122, HA-Ub (WT) and Flag-MDA5, Flag-MDA5(K23R), Flag-MDA5 (K43R), Flag-MDA5 (K68R), Flag-MDA5 (K128R), Flag-MDA5 (K137R), Flag-MDA5 (K169R), Flag-MDA5 (K174R), Flag-MDA5 (K240R), Flag-MDA5 (K68RK137R) or not. Flag Beads were used for CO-IP 24 h after transfection and further detected by Western blotting with an anti-HA, an anti-Flag antibody and an anti-Myc antibody, respectively. (**f**) HEK293T cells were co-transfected with Myc-RNF122, HA-Ub (K27) and Flag-MDA5, Flag-MDA5 (K68RK137R) or not, 24 h after transfection, Flag Beads were used for CO-IP, and further detected by Western blotting with an anti-HA, an anti-Flag antibody and an anti-Myc antibody, respectively. (**g**) HEK293T cells were co-transfected with Myc-RNF122, HA-Ub (K48) and Flag-MDA5, Flag-MDA5 (K68RK137R) or not, 24 h after transfection, Flag Beads were used for CO-IP, and further detected by Western blotting with an anti-HA, an anti-Flag antibody and an anti-Myc antibody, respectively.

**Table 1 viruses-14-00424-t001:** The primers used for PCR amplification.

Primer Name	Genbank Number	Sequence of Primer (5′-3′)
19T-RNF122-F	XM_013984096.2	GCGCGTTCCTTGTCAGTTTT
19T-RNF122-R		TTGCCACCCAACAGTCTTGT
pCMV-RNF122-F	XM_013984096.2	ccagtcgactctagaggatccATGCACCCATTTCAGTGGTGTA
pCMV-RNF122-R		cagggatgccacccgggatccTCACACCAGTTCATCCAGTAGAATC
pet-28a-RNF122-F	XM_013984096.2	cagcaaatgggtcgcggatccATGCACCCATTTCAGTGGTGTA
pet-28a-RNF122-R		acggagctcgaattcggatccTCACACCAGTTCATCCAGTAGAATC
Myc-RNF122(TM)-F	XM_013984096.2	ccagtcgactctagaggatccATGCACCCATTTCAGTGGTGTA
Myc-RNF122(TM) -R		cagggatgccacccgggatccTCATTTAAGCACCACCTCTTTATATCC
Myc-RNF122(RING)-F	XM_013984096.2	ccagtcgactctagaggatccATGGGTGATGCCAAGAAGTTACA
Myc-RNF122(RING) -R		cagggatgccacccgggatccTCACACCAGTTCATCCAGTAGAATC
Flag-MDA5-F	NM_001100194.1	ccagtcgactctagaggatccATGTCGTCGGATGGGTATTCC
Flag-MDA5-R		cagggatgccacccgggatccTCAGTCCTCATCACTAGACAAACAATAT
Flag-MDA5(N)-F	NM_001100194.1	ccagtcgactctagaggatccATGTCGTCGGATGGGTATTCC
Flag-MDA5(N)-R		cagggatgccacccgggatccTCAAGTCTCTTCATCTGAATCACTTCC
Flag-MDA5(M)-F	NM_001100194.1	ccagtcgactctagaggatccATGGTGGCTCAAAGAGCATCC
Flag-MDA5(M)-R		cagggatgccacccgggatccTCAGGTGCTCTCATCAGCTCTG
Flag-MDA5(C)-F	NM_001100194.1	ccagtcgactctagaggatccATGTACGTCCTGGTTGCCCA
Flag-MDA5(C)-R		cagggatgccacccgggatccTCAGTCCTCATCACTAGACAAACAATAT
Flag-MDA5(K23R)-F	NM_001100194.1	TGTTTCAGGGCCAGAGTGAGAAGGTACATT
Flag-MDA5(K23R)-R		CTCACTCTGGCCCTGAAACACGAGATGAGA
Flag-MDA5(K43R)-F	NM_001100194.1	TTTCTGCCTGCAGAGGTGAGGGAGCAGATT
Flag-MDA5(K43R)-R		CTCACCTCTGCAGGCAGAAAGGTCAAGTAG
Flag-MDA5(K68R)-F	NM_001100194.1	CTTCTGAACACTTTGGAGAGGGGGGTCTGG
Flag-MDA5(K68R)-R		CTCTCCAAAGTGTTCAGAAGCAGTTCAGCT
Flag-MDA5(K128R)-F	NM_001100194.1	CAGCCTACAGTGGTGGACAGGCTTCTGGTT
Flag-MDA5(K128R)-R		CTGTCCACCACTGTAGGCTGAAGAAGGTTC
Flag-MDA5(K137R)-F	NM_001100194.1	GTTACCGATGTCTTGGATAGATGTGTGGAG
Flag-MDA5(K137R)-R		CTATCCAAGACATCGGTAACCAGAAGCTTG
Flag-MDA5(K169R)-F	NM_001100194.1	GGAGTAAGGGAGCTCCTGAGAAGGATTGTG
Flag-MDA5(K169R)-R		CTCAGGAGCTCCCTTACTCCTGATTCATTT
Flag-MDA5(K174R)-F	NM_001100194.1	CTGAAAAGGATTGTGCAGAGAGAAAACTGG
Flag-MDA5(K174R)-R		CTCTGCACAATCCTTTTCAGGAGCTCCCTT
Flag-MDA5(K240R)-F	NM_001100194.1	GACGTCTCGGACATAGAGAGAAGTTCACTG
Flag-MDA5(K240R)-R		CTCTCTATGTCCGAGACGTCCAGACTTGGC
Flag-nsp1α-F	JX317649.1	caagcttgcggccgcgaattcaATGTCTGGGATACTTGATCGGTG
Flag-nsp1α-R		cagggatgccacccgggatccTCAAGCACACTCAAAAGGGCA
Flag-nsp1β-F	JX317649.1	caagcttgcggccgcgaattcaATGGCTGACGTCTATGACATTGG
Flag-nsp1β-R		cagggatgccacccgggatccTCAACCGTACCACTTATGACTGCC
Flag-nsp4-F	JX317649.1	aagcttgcggccgcgaattcaATGGGCGCTTTCAGAACTCA
Flag-nsp4-R		cagggatgccacccgggatccTCATTCCAGTTCGGGTTTGG
Flag-nsp5-F	JX317649.1	caagcttgcggccgcgaattcaATGGGAGGCCTTTCCACAGT
Flag-nsp5-R		cagggatgccacccgggatccTCACTCGGCAAAGTATCGCA
Flag-nsp7-F	JX317649.1	caagcttgcggccgcgaattcaATGTCGCTGACTGGTGCCC
Flag-nsp7-R		cagggatgccacccgggatccTCATTCCCACTGAGCTCTTCTATTC
Flag-nsp9-F	JX317649.1	caagcttgcggccgcgaattcaATGTTTAAACTGCTAGCCGCCA
Flag-nsp9-R		cagggatgccacccgggatccTCACTCATGATTGGACCTGAGTTT
Flag-nsp10-F	JX317649.1	caagcttgcggccgcgaattcaATGGGGAAGAAGTCCAGAATGTG
Flag-nsp10-R		cagggatgccacccgggatccTCATTCCAGGTCTGCGCAA
Flag-nsp11-F	JX317649.1	caagcttgcggccgcgaattcaATGGGGTCGAGCTCCCCG
Flag-nsp11-R		cagggatgccacccgggatccTCATTCAAGTTGAAAATAGGCCG
Flag-N-F	JX317649.1	caagcttgcggccgcgaattcaATGCCAAATAACAACGGCAAG
Flag-N-R		cagggatgccacccgggatccTCATGCTGAGGGTGATGCTGT
Flag-nsp4(K7R)-F	JX317649.1	GGCGCTTTCAGAACTCAAAGGCCCTCACTG
Flag-nsp4(K7R)-R		CTTTGAGTTCTGAAAGCGCCCATGAATTCG
Flag-nsp4(K33R)-F	JX317649.1	ACTATTGACGGGAAAATCAGGTGCGTGACT
Flag-nsp4(K33R)-R		CTGATTTTCCCGTCAATAGTGAACACTCCG
Flag-nsp4(K79R)-F	JX317649.1	TGGCAAGGGGTTGCTCCCAGGGCCCAGTTC
Flag-nsp4(K79R)-R		CTGGGAGCAACCCCTTGCCAATTCGGGCAA
Flag-nsp4(K158R)-F	JX317649.1	TGTAATGTGAAGCCCATCAGGCTGAGCGAG
Flag-nsp4(K158R)-R		CTGATGGGCTTCACATTACAAAACTGGCCT
Flag-nsp4(K170R)-F	JX317649.1	GAATTCTTCGCTGGACCTAGGGTCCCGCTC
Flag-nsp4(K170R)-R		CTAGGTCCAGCGAAGAATTCACTCAACTCG

**Table 2 viruses-14-00424-t002:** Primers used in small RNA interfering assay.

Primer Name	Primer Sequence (5′-3′)
negative control	F:UUCUCCGAACGUGUCACGUTT
	R:ACGUGACACGUUCGGAGAATT
siRNF122-1	F: CCAUGCCACCCAUCAGUUUTT
	R:AAACUGAUGGGUGGCAUGGTT
siRBF122-2	F: GGACGAGCUAGGUGUGCUUTT
	R: AAGCACACCUAGCUCGUCCTT

**Table 3 viruses-14-00424-t003:** The primers used for Luciferase reporter gene.

Primer Name	Genbank Number	Sequence of Primer (5′-3′)
pTP1-Luc-F	NC_010457.5	ggggtaccGGGATTGAACCCACATCCACA
pTP1-Luc-R		ccgctcgagACCACTGAAATGGGTGCATCA
pTP2-Luc-F	NC_010457.5	ATGGATGGATGTTCCTGGCATCTCACCTCC
pTP2-Luc-R		ATCCATCCATGTGGATGTGGGTTCAATCCC
pTP3-Luc-F	NC_010457.5	ATGGATGGATGGTCCCCAGGCTAGGGGTCC
pTP3-Luc-R		ATCCATCCATGTGGATGTGGGTTCAATCCC
pTP4-Luc-F	NC_010457.5	ATGGATGGATGCCAGATCCTTAACCCACTG
pTP4-Luc-R		ATCCATCCATGTGGATGTGGGTTCAATCCC
pTP5-Luc-F	NC_010457.5	ATGGATGGATTCTCTAGGTCCATCCCCCTT
pTP5-Luc-R		ATCCATCCATGTGGATGTGGGTTCAATCCC
pTP6-Luc-F	NC_010457.5	ATGGATGGATTGCGTGCTGGCCGGTAAATA
pTP6-Luc-R		ATCCATCCATGTGGATGTGGGTTCAATCCC
pTP7-Luc-F	NC_010457.5	ATGGATGGATGAGCTTCCCGGGGAGAGGGG
pTP7-Luc-R		ATCCATCCATGTGGATGTGGGTTCAATCCC
pTP8-Luc-F	NC_010457.5	ATGGATGGATGTTCCTGGCATCTCACCTCC
pTP8-Luc-R		ATCCATCCATGTGGATGTGGGTTCAATCCC
pTP9-Luc-F	NC_010457.5	ATGGATGGATCTGCCTTTCCTCGCCGTGGT
pTP9-Luc-R		ATCCATCCATGTGGATGTGGGTTCAATCCC
pTP6-M1-Luc-F	NC_010457.5	CTCCCATTGGCTGGAGGGAAGGAGACGGGC
pTP6-M1-Luc-R		TTCCCTCCAGCCAATGGGAGCCTTCCTACG
pTP6-M2-Luc-F	NC_010457.5	AAATAATACCCGGGGCCCAGCGGAAGGCTC
pTP6-M2-Luc-R		GCTGGGCCCCGGGTATTATTTACCGGCCAG
pTP6-M3-Luc-F	NC_010457.5	CCGTAGGAAGGCTCCCATCAACTGGAGGGC
pTP6-M3-Luc-R		TTGATGGGAGCCTTCCTACGGGCCCCGGGT

**Table 4 viruses-14-00424-t004:** The primers used for qRT-PCR amplification.

Primer Name	Genbank Number	Sequence of Primer (5′-3′)
RNF122-F	XM_013984096.2	ACATGGTCATCTTCGGCACA
RNF122-R		AGACTGCACAGGTCCCGTA
PRRSV-N-F	ABR37297.1	CAGTCAATCAGCTGTGCCAAA
PRRSV-N-R		ATCTGACAGGGCACAAGTTCCA
PRRSV-nsp2-F	ABR37297.1	CAGCCTTATGACCCCAACCAG
PRRSV-nsp2-R		TGGGCAAAGTCCCCTGTACCAA
IFNβ-F	NM_001003923	GCAGTATTGATTATCCACGAGA
IFNβ-R		TCTGCCCATCAAGTTCCAC
NF-κB -F	X61498.1	CCCAGCCATTTGCACACCTCAC
NF-κB -R		TTCAGAATTGCCCGACCAGTTTTT
β-actin-F	DQ452569.1	GAATCCTGCGGCATCCACGA
β-actin-R		CTCGTCGTACTCCTGCTTGCT
RB-F	XM_013992198.2	TCTCCTTTAAGATCCCCCAAGAA
RB-R		TTGAGGTTGCTTGTGCCTCT
E2F1-F	XM_021077692.1	CGGCTTGAAGGATTGACCCA
E2F1-R		TCAGCATCCTCGGAAAGCAG
E2F4-F	XM_003126933.6	AACGTGCTGGAAGGTATCGG
E2F4-R		CTTGTCCGCAATTTCCCGTG
E2F6-F	XM_005655271.3	GTTGGATGTTCCTGCTCCCA
E2F6-R		CCGTCCGACACTTTACTGCT
FOXC1-F	XM_005665529.3	GATGTTCGAGTCGCAGAGGAT
FOXC1-R		CAGAACTTGCTGCAGTCGTAG
NFIC-F	XM_021084124.1	GGATGTATTCGTCCCCGCTC
NFIC-R		GTTGAACCAGGTGTAGGCGA
HLTF-F	XM_013991989.2	TCGTGTTAGAGACCCAGCCT
HLTF-R		TCCAGGATCACTCTTAGCCAC

**Table 5 viruses-14-00424-t005:** Transcription factor binding sites of RNF122.

Model ID	Model Name	Score	Relative Score	Start	End	Strand	Predicted Site Sequence
MA0032.1	FOXC1	5.723	0.902187783821075	28	35	1	GGCCCGTA
MA0109.1	HLTF	6.251	0.901528565109457	43	52	1	TCCCATTGGC
MA0161.1	NFIC	8.520	0.960563817861358	48	53	1	TTGGCT
MA0024.2	E2F1	14.481	0.985884368151812	56	66	1	AGGGCGGGAGA
MA0470.1	E2F4	14.473	0.974005560828083	57	67	1	GGGCGGGAGAC
MA0471.1	E2F6	14.993	0.9822249281057	57	67	1	GGGCGGGAGAC
MA0470.1	E2F4	10.088	0.904564545543506	68	78	1	GGGCGGGGGGC

**Table 6 viruses-14-00424-t006:** The type I interferon signaling pathway associated proteins in the RNF122 interaction proteins were identified by mass spectrometry.

Description	Mass	Score
RING finger protein 122 (RNF122)	18,204	1381
Transitional endoplasmic reticulum ATPase (VCP)	89,950	93
Zinc finger CCCH-type antiviral protein 1 (ZC3HAV1)	103,135	93
TNFAIP3-interacting protein 2 (TNIP2)	49,240	33
Non-receptor tyrosine-protein kinase TYK2 (TYK2)	135,389	31
Interferon-induced helicase C domain-containing protein 1 (MDA5)	117,926	21
DNA-directed RNA polymerase III subunit RPC2 (POLR3B)	129,242	20
E3 ubiquitin-protein ligase TRIM69 (TRIM69)	58,351	17

## Data Availability

The data presented in this study are available on request from the corresponding author.

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
