# Peer review of "PRRSV Non-Structural Proteins Orchestrate Porcine E3 Ubiquitin Ligase RNF122 to Promote PRRSV Proliferation"

_viruses, 2022, doi:10.3390/v14020424_

Round 1

Reviewer 1 Report

Sun and coworkers report RNF122 is upregulated during PRRSV infection and promotes PRRSV infection. The authors have found the promoter of RNF122 and identified transcription factors responsible for regulation. Further nsp1α, nsp7 and nps9 inhibit the function of transcription factors HLTF and E2F complex respectively, which in turn promotes RNF122 upregulation. RNF122 inhibits MDA5 and enhance stabilization of nsp4 by ubiquitination of K27, K48 and K63 respectively, where both promoted PRRV virus replication. They observed increased mRNA level of nsp2, N protein of PRRSV which indicates increased replication. RNF122 negatively regulated type I IFN signaling. The idea of the topic is novel and study show very interesting results. The data is well presented in most part but needs some improvement in the following areas.  

Note to authors to check the following under the sections: 

3.1  

  • The role of RNF122 was not explained in the introduction or discussion.  
  • It was not clearly mentioned or reference is made for the use of PCV2 in the experiment. Negative control may have been included in the experiment  
  • Why the authors did not look at the protein level of RNF122 after the PRRSV infection? 

3.2. 

  • Can author explain any software was used for finding the potent promoter sequence for RNF122 clearly? 

3.4. 

  • Fig 4d and 4e: have you looked at combined effect of nsp1, nsp7 and checked nsp9 effect? 
  • In Fig 4 what is the purpose of western blot image. I see no explanation made in the text about the figure? Any reason for the varied expression of different proteins? 

3.5. 

  • Fig 5g and 5h: Did the authors performed statistical analysis and found statistical significance of virus replication experiment? Why there is one point plotted in fig 5g compared to 5h. 
  • Some blots not developed very well to arrive at a final conclusion of the findings for ex. Fig 5c, 5f, 5i. 
  • Which gene or sequences were used for viral copies measurement of PRRSV is not explained clearly in the methods section? 
  • Fig 5i experiment not discussed.  
  • Table 7 is missing from the manuscript. 

3.6. 

  • Fig 6b. Merge spelling mistake 
  • Fig 6c: how do you explain that no myc-RNF122 were visible in the WB but still it strengthens nsp4? 
  • Although nsp4 function in PRRSV replication, authors concluded nsp4 promoted PRRSV replication. Did the authors performed virology experiment to determine the effect of nsp4? 

3.8. 

  • I do not see any lane markers in original blots. Why it is missing? 
  • Did the authors considered looking at IRF-3 levels? 
  • Some blots not developed very well to arrive at a final conclusion of the findings for ex. Fig 8c. 

  • qRT-PCR, ubiquitination assay methods are not explained well or reference not made.  

There are inaccurate sentences in the following lines. 

L.112 pTP1-Luc Fireflies or firefly? 

L.178 as our previously described. The total of 26 significantly changed   

L.220 of interested in genes  

L.233-4 in regulation RNF122 expression 

L.244 Sentence have problem: This suggests that PRRSV can further release the activity of E2F complex and promote the transcription of RNF122 by down-regulating the transcription level of RB. 

L.260 Sentence have problem: performed co-transfected with pTP6-Luc, respectively.  

L.272-273 Sentence have problem: We need clearer and more definite whether nsp1α and nsp7 impact the transcription of RNF122 by regulating the expression of HLTF.  

L.280 Use correct term for transfection of DNA: infected with PRRSV (MOI=0.5) nsp1α,nsp4,nsp5, nsp7, nsp9, nsp10. 

L.283 Transfected is the right term: infected with nsp9 or not 

L.301 RNA interference of gene is not mentioned?  

L.320 Why the word used: interaction ships? 

L.323 Gene should be included in the sentences: different PRRSV encoding plasmids. 

L.344 Sentence not clear: five lysine mutants of nsp4 were constructed according to be described in references and co-transfected with RNF122. 

L.462 Sentence can be improved: PRRSV inhibits effects of the type I IFN signaling pathway can be performed by viral encoding proteins, such as nsp1α, nsp1β, nsp2, nsp4, nsp11, and N. 

I have found many spelling errors and it should be addressed later. 

Author Response

Dear Reviewer,

Thanks for you careful reading, thoughtful comments and valuable suggestions on our manuscript. We have modified our manuscript and supplemented some figures according the your suggestions as follows. The English grammatical mistakes, words and phrases have been corrected and polished in our update manuscript. We look forward to your positive response.

Best regards

Jinhai Huang

Sun and coworkers report RNF122 is upregulated during PRRSV infection and promotes PRRSV infection. The authors have found the promoter of RNF122 and identified transcription factors responsible for regulation. Further nsp1α, nsp7 and nps9 inhibit the function of transcription factors HLTF and E2F complex respectively, which in turn promotes RNF122 upregulation. RNF122 inhibits MDA5 and enhance stabilization of nsp4 by ubiquitination of K27, K48 and K63 respectively, where both promoted PRRV virus replication. They observed increased mRNA level of nsp2, N protein of PRRSV which indicates increased replication. RNF122 negatively regulated type I IFN signaling. The idea of the topic is novel and study show very interesting results. The data is well presented in most part but needs some improvement in the following areas. 

Note to authors to check the following under the sections:

Point 1: The role of RNF122 was not explained in the introduction or discussion. It was not clearly mentioned or reference is made for the use of PCV2 in the experiment. Negative control may have been included in the experiment. Why the authors did not look at the protein level of RNF122 after the PRRSV infection?

Response 1: Thanks for your valuable suggestions. We have supplemented the relevant background of RNF122 in the introduction and discussion. According to the reviewer’s suggestion, we have deleted the relevant contents between PCV2 and RNF122. And the protein level of RNF122 after the PRRSV infection was also supplemented in Fig 1c.

Point 2: Can author explain any software was used for finding the potent promoter sequence for RNF122 clearly?

Response 2: Thanks for your comments. The potent promoter sequences of the porcine RNF122 gene initial codons were predicted based on the porcine genomic sequences analysis (https://www.ncbi.nlm.nih.gov/gene/100511414). In addition, we synthesized the prediction results from the Promoter2.0(https://services.healthtech.dtu.dk/service.php?Promoter-2.0), NNPP(http://fruitfly.org:9005/segtools/promoter.html), and SoftBerry(http://www.softberry.com/berry.phtml?topic=index&group=programs&subgroup=promoter) softwares, and then designed and constructed 8 truncated mutants to further verify the region of the core promoter. We have supplemented the statements in our manuscript.

Point 3: Fig 4d and 4e: have you looked at combined effect of nsp1, nsp7 and checked nsp9 effect? In Fig 4 what is the purpose of western blot image. I see no explanation made in the text about the figure? Any reason for the varied expression of different proteins?

Response 3: Thanks for your valuable comments. We wanted to determine which PRRSV coding protein had an effect on the transcription factors of RNF122, it is a good suggestions to further evaluate the combined effect. The western blot image in Fig 4 correlates to Fig 4a, represents the expression situation of the transfection plasmids, We consider that the different expression of proteins may be related to their properties.

Point 4: Fig 5g and 5h: Did the authors performed statistical analysis and found statistical significance of virus replication experiment? Why there is one point plotted in fig 5g compared to 5h. Some blots not developed very well to arrive at a final conclusion of the findings for ex. Fig 5c, 5f, 5i. Which gene or sequences were used for viral copies measurement of PRRSV is not explained clearly in the methods section? Fig 5i experiment not discussed. Table 7 is missing from the manuscript.

Response 4: Thanks for your comments. We performed statistical analysis with three replicates to demonstrate significant differences between the experimental groups. The points in Fig 5g and 5h are different only in the way of plotting, the Fig 5h gave triplicate detailed results .

The effect of RNF122 on PRRSV replication was confirmed by Fig 5a, 5b, 5d-5h, and for the accuracy of the experiment, we performed western blotting and quantified by Fig 5c, 5f, 5i. N gene was used for viral copies measurement of PRRSV, we have added the supplementary notes in the legend. Fig 5i represents the overexpression and interference expression of RNF122, in order to confirm the reliability of the experimental group set in Fig 5g and 5h. we have rearranged the order of the tables in the manuscript to make it more clearly.

Point 5: Fig 6b. Merge spelling mistake. Fig 6c: how do you explain that no myc-RNF122 were visible in the WB but still it strengthens nsp4? Although nsp4 function in PRRSV replication, authors concluded nsp4 promoted PRRSV replication. Did the authors performed virology experiment to determine the effect of nsp4?

Response 5: Thank you for the carefully reading and suggestions.  We have corrected the misspelling.

In Fig 6c, the second band has little myc-RNF122 (0.1ug) added, and it may not be displayed clearly under the same exposure conditions. Fig 5g and 5h can prove that the RNF122 promotes virus proliferation, and RNF122 can stabilize the expression of viral protein nsp4. Therefore, we speculate that there may be the possibility that RNF122 indirectly promotes virus replication by stabilizing viral proteins.

Point 6: I do not see any lane markers in original blots. Why it is missing? Did the authors considered looking at IRF-3 levels? Some blots not developed very well to arrive at a final conclusion of the findings for ex. Fig 8c.

Response 6: Thanks for your comments. As some images need to be cut and then exposed, not all the original images have noted markers. We detected several indicators of the interferon pathway, and we mainly wanted to focus on signaling molecules in the upstream pathway,so only some key point , such as the IFN-β and NF-κB were detected. The Fig 8c and 8g are used to prove the expression of RNF122 at the protein level, so as to ensure the reliability of the experimental results in Fig 8.

Point 7: qRT-PCR, ubiquitination assay methods are not explained well or reference not made. 

Response 7: Thanks for your valuable suggestions. We have supplemented the relevant content in 2.6 and 2.7.

There are inaccurate sentences in the following lines.

Point 8: L.112 pTP1-Luc Fireflies or firefly?

Response 8: Thank you for the carefully reading and suggestions. We corrected Fireflies to Firefly.

Point 9: L.178 as our previously described. The total of 26 significantly changed  

Response 9: Thank you for the carefully reading and suggestions. We have adjusted the sentence. "A total of 26 homologous enzymes containing RING domain E3 ubiquitin ligases in the transcriptomic data were further identified and the significantly up-regulated RNF122 was screened for further study. (Fig.1a). "

Point 10: L.220 of interested in genes 

Response 10: Thank you for the carefully reading and suggestions. We have deleted "interested in".

Point 11: L.233-4 in regulation RNF122 expression

Response 11: Thank you for the carefully reading and suggestions. We have corrected it to "regulating".

Point 12: L.244 Sentence have problem: This suggests that PRRSV can further release the activity of E2F complex and promote the transcription of RNF122 by down-regulating the transcription level of RB.

Response 12: Thank you for the carefully reading and suggestions. We have adjusted the sentence. “This suggested that PRRSV can further reduce the activity of E2F complex and promote the transcription of RNF122 by down-regulating the transcription level of RB”.

Point 13: L.260 Sentence have problem: performed co-transfected with pTP6-Luc, respectively. 

Response 13: Thanks for your valuable comments. We have corrected the sentence.

Point 14: L.272-273 Sentence have problem: We need clearer and more definite whether nsp1α and nsp7 impact the transcription of RNF122 by regulating the expression of HLTF. 

Response 14: Thanks for your valuable comments. We have corrected the sentence. “We need to define whether nsp1α and nsp7 impact the transcriptional level of RNF122 by regulating the expression of HLTF.”

Point 15: L.280 Use correct term for transfection of DNA: infected with PRRSV (MOI=0.5) nsp1α, nsp4, nsp5, nsp7, nsp9, nsp10.

Response 15: Thanks for your valuable comments. We have corrected the sentence. “HEK293T cells were co-transfected with pTP6-Luc and pRL-TK, and then transfected nsp1α, nsp1β, nsp4, nsp5, nsp7, nsp9, nsp10, nsp11, N plasmids respectively, or infected with PRRSV (MOI=0.5).”

Point 16: L.283 Transfected is the right term: infected with nsp9 or not

Response 16: Thank you for the carefully reading and suggestions. We have corrected the "infected" to the "transfected".

Point 17: L.301 RNA interference of gene is not mentioned? 

Response 17: Thank you for the carefully reading and suggestions. We have added “RNF122 siRNA”

Point 18: L.320 Why the word used: interaction ships?

Response 18: Thank you for the carefully reading and suggestions. We have deleted " ships ".

Point 19: L.323 Gene should be included in the sentences: different PRRSV encoding plasmids.

Response 19: Thanks for your valuable comments. We have corrected the sentence. “RNF122 plasmid was co-transfected with expression vectors that encode PRRSV proteins.”

Point 20: L.344 Sentence not clear: five lysine mutants of nsp4 were constructed according to be described in references and co-transfected with RNF122.

Response 20: Thanks for your valuable comments. We have adjusted the sentence. “five lysine mutants of nsp4 were constructed based on references and co-transfected with RNF122.”

Point 21: L.462 Sentence can be improved: PRRSV inhibits effects of the type I IFN signaling pathway can be performed by viral encoding proteins, such as nsp1α, nsp1β, nsp2, nsp4, nsp11, and N.

Response 21:Thanks for your comments. We have adjusted the sentence.“PRRSV encodes several proteins, such as nsp1α, nsp1β, nsp2, nsp4, nsp11, and N that act as antagonists for the IFN signaling”

I have found many spelling errors and it should be addressed later.

Thanks for your valuable comments. We have revised the manuscript with the help of a native English-speaking expert in my field.

Reviewer 2 Report

Summary: The authors demonstrate that PRRSV non-structural proteins manipulate the expression and stability of porcine RNF122 for productive viral replication. Results show that multiple viral non-structural proteins are needed to regulate transcription factor stability and binding at the RNF122 promoter site. The authors also demonstrate the nsp4-dependent association with RNF122 and specific ubiquitination, and degradation, of MAD5 needed to suppress type 1 interferon response to PRRSV infection. The manuscript is an important contribution to the PRRSV field and I would suggest accepting the manuscript after revisions.

Major Comments:

  1. Many of the western blots would benefit from a quantitative measurement of the detected signal. For example, in Figure 4E nsp1α expression is barely detectable and thus it appears to have very little impact on the expression of HTLF. Quantification of the signal on the blot might help with interpreting the results, particularly as the expression of transfected proteins varies.

  1. Figure 7a: Both the β-actin loading control and, more importantly, the amount of RNF122 in the input for Figure 7a are much higher in the co-IP with the K63 plasmid. This may be influencing the results. Can the authors comment on this in the manuscript or provide a co-IP with more even loading?

  1. Line 492-494: The authors suggest that RNF122-mediated K63 ubiquitination of PRRSV nsp4 stabilizes the nsp4 protein expression. This was not demonstrated in Figure 7a where RNF122 co-transfection appears to result in decreased nsp4 levels (see IB: Flag in the IP fraction) or 7b where a siRNA-induced knockdown of RNF122 does not appear to impact the level of nsp4. Similarly, in supplemental Figure 1B, the level of nsp4 doesn’t change, despite the K63 begin show to be specifically linked to nsp4. Can the authors comment on this or think of an experiment to directly test whether nsp4 stability is impacted by K63 ubiquitination?

  1. Figure 4: The authors demonstrate that nsp9 promotes RNF122 expression by suppressing RB at the transcriptional level, corroborating previous data showing nsp9-dependent degradation of RB. While the authors also indicate that nsp1α and 7 also impact expression from the RNF122 promoter (Figure 4A), they only test the impact of these proteins on HTLF expression, but never look at the impact of these proteins on RB. Can the authors comment on this? Similarly, does nsp9 impact HTLF?

  1. Figure 6A: While the authors clearly demonstrate the interaction between PRRSV nsp4 and RNF122, there is at least one weak, but detectable interaction between nsp7 and RNF122 that the authors don’t address. Also, considering the expression levels vary greatly from one PRRSV nsp to the next (see both the input and IP fraction western blots probed with FLAG antibody, particularly for nsp7), there may be other interactions we are not able to observe.

Minor Comments:

  1. The text in the figures is often varying in size, some of which are very difficult to read and have poor resolution (e.g. labeling on Fig. 2a, 4b western blot, Fig 7c, etc.). Label sizes should be easier to read and the resolution checked.
  2. Line 237: “….from missing detection signal of NFIC…” NFIC was not listed on this graph. It would be appropriate to list NFIC on the graph and simply mark it as N.D. (Not Detected).
  3. Consider relabeling the tables in sequential order with the first table shown (Table 4 primers used for PCR amplification) being Table 1. Relabeling the tables may also explain why the siRNA sequences, introduced in the methods as Table 5 (line 107), are found in Table 2 (pg 11) in the results. In contrast, Table 5 contains the primers used in the luciferase assays.
  4. Line 196: Should read as RNF122
  5. Line 204: Should read as “Fig. 2a” not “Fig. 2c”
  6. Line 299: Should read as “Fig. 5a and b” not just “Fig. 5a.” Also, at the end of this sentence, “Fig. 5b” should read as “Fig. 5c.”
  7. Figure 5i is never mentioned in the results and needs to be properly labeled with which lane corresponds to PRRSV infection, which was co-transfected with siRNA, etc.
  8. Figure 4a and 5a: Consider labeling the specific bands corresponding to the various nsp proteins expressed. This is particularly pertinent for Figure 4 where the ~18.7 kDa size nsp5 is not even detectable on Figure 4a.

Author Response

Dear Reviewer,

Thanks for you careful reading, thoughtful comments and valuable suggestions on our manuscript. We have modified our manuscript and supplemented some figures according the your suggestions as follows. The English grammatical mistakes, words and phrases have been corrected and polished in our update manuscript. We look forward to your positive response.

Best regards

Jinhai Huang

The authors demonstrate that PRRSV non-structural proteins manipulate the expression and stability of porcine RNF122 for productive viral replication. Results show that multiple viral non-structural proteins are needed to regulate transcription factor stability and binding at the RNF122 promoter site. The authors also demonstrate the nsp4-dependent association with RNF122 and specific ubiquitination, and degradation, of MAD5 needed to suppress type 1 interferon response to PRRSV infection. The manuscript is an important contribution to the PRRSV field and I would suggest accepting the manuscript after revisions.

Major Comments:

Point 1: Many of the western blots would benefit from a quantitative measurement of the detected signal. For example, in Figure 4E nsp1α expression is barely detectable and thus it appears to have very little impact on the expression of HTLF. Quantification of the signal on the blot might help with interpreting the results, particularly as the expression of transfected proteins varies.

Response 1: Thanks for your valuable comments. As you say, it is difficult to get all protein expression result when transfected different plasmids into cells, owing to variable expresion and stable status of those protein. Here, we tried to evaluate them by different experiments to obtain more reliable result.

Point 2: Figure 7a: Both the β-actin loading control and, more importantly, the amount of RNF122 in the input for Figure 7a are much higher in the co-IP with the K63 plasmid. This may be influencing the results. Can the authors comment on this in the manuscript or provide a co-IP with more even loading?

Response 2: Thanks for your careful reading and valuable comments. In Fig 7a, there is no comparison between the two groups of WT and K63, and which should be compared within the group separately, such as the first band is compared with the second band, and the third band is compared with the fourth band. Therefore, we believe that the difference in cell volume between WT and K63 groups have little impact to our conclusion. 

Point 3: Line 492-494: The authors suggest that RNF122-mediated K63 ubiquitination of PRRSV nsp4 stabilizes the nsp4 protein expression. This was not demonstrated in Figure 7a where RNF122 co-transfection appears to result in decreased nsp4 levels (see IB: Flag in the IP fraction) or 7b where a siRNA-induced knockdown of RNF122 does not appear to impact the level of nsp4. Similarly, in supplemental Figure 1B, the level of nsp4 doesn’t change, despite the K63 begin show to be specifically linked to nsp4. Can the authors comment on this or think of an experiment to directly test whether nsp4 stability is impacted by K63 ubiquitination?

Response 3: Thanks for your valuable comments. RNF122 promotes stable expression of nsp4 was demonstrated by Fig 6c. We believe that RNF122 should not affect the IP level of nsp4 in the CO-IP experiment. Fig 7a represents that RNF122 performs K63-linked ubiquitination of PRRSV nsp4 and the level of flag in IP should be consistent. The decreased nsp4 levels may be contribute to the error caused by our operation or imaging process.

Point 4: Figure 4: The authors demonstrate that nsp9 promotes RNF122 expression by suppressing RB at the transcriptional level, corroborating previous data showing nsp9-dependent degradation of RB. While the authors also indicate that nsp1α and 7 also impact expression from the RNF122 promoter (Figure 4A), they only test the impact of these proteins on HTLF expression, but never look at the impact of these proteins on RB. Can the authors comment on this? Similarly, does nsp9 impact HTLF?

Response 4: Thanks for your comments. It has been reported that the nsp9 has an effect on RB. Therefore, we did not consider exploring the influence of other PRRSV-encoded proteins on RB, here we focus on the effects of nsp1α and nsp7 on another transcription factor HLTF.

Point 5: Figure 6A: While the authors clearly demonstrate the interaction between PRRSV nsp4 and RNF122, there is at least one weak, but detectable interaction between nsp7 and RNF122 that the authors don’t address. Also, considering the expression levels vary greatly from one PRRSV nsp to the next (see both the input and IP fraction western blots probed with FLAG antibody, particularly for nsp7), there may be other interactions we are not able to observe.

Response 5: Thanks for your valuable comments. In Fig 6a, the molecular weight of Myc-RNF122 is 27kDa in IP group, which is also the blot of our main detection. Only the nsp4 group has obvious bands, which is indicated by arrow, and the others are non-specific bands in each channel.

Minor Comments:

Point 1: The text in the figures is often varying in size, some of which are very difficult to read and have poor resolution (e.g. labeling on Fig. 2a, 4b western blot, Fig 7c, etc.). Label sizes should be easier to read and the resolution checked.

Response 1:Thanks for your valuable comments. We have adjusted the relevant figures.

Point 2: Line 237: “….from missing detection signal of NFIC…” NFIC was not listed on this graph. It would be appropriate to list NFIC on the graph and simply mark it as N.D. (Not Detected). 

Response 2: Thanks for your valuable comments. We have listed NFIC on the graph.

Point 3: Consider relabeling the tables in sequential order with the first table shown (Table 4 primers used for PCR amplification) being Table 1. Relabeling the tables may also explain why the siRNA sequences, introduced in the methods as Table 5 (line 107), are found in Table 2 (pg 11) in the results. In contrast, Table 5 contains the primers used in the luciferase assays.

Response 3: Thanks for your valuable suggestions. We have rearranged the order of the tables in the manuscript.

Point 4: Line 196: Should read as RNF122

Response 4: Thank you for the carefully reading and suggestions. We have corrected the misspelling.

Point 5: Line 204: Should read as “Fig. 2a” not “Fig. 2c”

Response 5: Thanks for your comments. Fig 2a is a schematic diagram of constitution of luciferase mutant plasmid. The enhanced promoter activity after PRRSV infection was demonstrated by Figure 2c. So here corresponds to Figure 2c.

Point 6: Line 299: Should read as “Fig. 5a and b” not just “Fig. 5a.” Also, at the end of this sentence, “Fig. 5b” should read as “Fig. 5c.” 

Response 6: Thanks for your valuable comments. We have adjusted the relevant content.

Point 7: Figure 5i is never mentioned in the results and needs to be properly labeled with which lane corresponds to PRRSV infection, which was co-transfected with siRNA, etc.

Response 7: Thanks for your valuable comments. We have already marked. Fig 5i represents the overexpression and interference expression of RNF122, in order to confirm the reliability of the experimental group set in Fig 5g and 5h.

Point 8: Figure 4a and 5a: Consider labeling the specific bands corresponding to the various nsp proteins expressed. This is particularly pertinent for Figure 4 where the ~18.7 kDa size nsp5 is not even detectable on Figure 4a. 

Response 8: Thank you for the carefully reading and suggestions. Referring to the results of the IP group in Figure 6a, nsp5 can be significantly enriched, which proves that it can be expressed normally without affecting its function in cells. In Fig 4a, it may be because its expression quantity is relatively low and it has not been exposed.  

Round 2

Reviewer 1 Report

The comment were addressed.

Author Response

Dear Reviewer,

Thanks for you careful reading, thoughtful comments and valuable suggestions on our manuscript. The English grammatical mistakes, words and phrases have been corrected and polished in our update manuscript. We look forward to your positive response.

Best regards

Jinhai Huang

Reviewer 2 Report

The authors were able to address experimental concerns in the paper but there are still a number of text/editorial changes that are needed to improve the clarity of the manuscript. Previously only a few examples were listed and it seems that only these were edited. I would highly suggest the authors review the manuscript thoroughly as there are multiple examples where edits are needed. The following are just some of the examples that were not corrected in the previous draft or have been introduced in the new draft:

Line 36: Remove the "." from the end of reference 6 and 7 as this is a continuing sentence.

Line 58: "not only" is repeated twice and should read "not only target cellular proteins"

Line 70: "Here, We" should read "Here, we"

Line 134: Should read "Total RNA from the cells was extracted using the RNAiso Plus reagent purchased...."

Line 222 (and in other figures): "Data are presented to three independent experiments." Is the data shown representative or an average of three independent experiments?

Line 225: "To understand which molecular promoting the upregulation of RNF122...." There appears to be a word missing here.
